# Prevalent Mycotoxins in Animal Feed: Occurrence and Analytical Methods

**DOI:** 10.3390/toxins11050290

**Published:** 2019-05-22

**Authors:** Carolina Santos Pereira, Sara C. Cunha, José O. Fernandes

**Affiliations:** LAQV-REQUIMTE, Laboratory of Bromatology and Hydrology, Faculty of Pharmacy, University of Porto, Rua Jorge Viterbo Ferreira, n° 228, 4050-313 Porto, Portugal

**Keywords:** mycotoxins, feed, fungi, occurrence, analytical methods, contaminants

## Abstract

Today, we have been witnessing a steady tendency in the increase of global demand for maize, wheat, soybeans, and their products due to the steady growth and strengthening of the livestock industry. Thus, animal feed safety has gradually become more important, with mycotoxins representing one of the most significant hazards. Mycotoxins comprise different classes of secondary metabolites of molds. With regard to animal feed, aflatoxins, fumonisins, ochratoxins, trichothecenes, and zearalenone are the more prevalent ones. In this review, several constraints posed by these contaminants at economical and commercial levels will be discussed, along with the legislation established in the European Union to restrict mycotoxins levels in animal feed. In addition, the occurrence of legislated mycotoxins in raw materials and their by-products for the feeds of interest, as well as in the feeds, will be reviewed. Finally, an overview of the different sample pretreatment and detection techniques reported for mycotoxin analysis will be presented, the main weaknesses of current methods will be highlighted.

## 1. Introduction

Feed is described by the European Commission as any substance or product, including additives, whether processed, semi-processed or unprocessed, intended to be used for oral feeding of animals [1]. It can be classified into the following four groups [2]:Forages—silage made from grass or cereal crops;Cereals and other home-grown crops—feeds with a high energy and/or protein content;Compound feeds—manufactured mixtures of single feed materials, minerals, and vitamins;Products and by-products of the human food and brewing industries—residues of vegetable processing, spent grains from brewing and malting and by-products of the baking, bread-making, and confectionery industries.

Livestock diets typically include a combination of feeds that are designed to meet not only the nutritional needs of animals with minimal costs, but also to provide everything they need for their health, welfare, and production [2,3]. However, cereals and cereal-based products are possibly the most commonly used ingredients in animal feed, supplying most of the nutrients for livestock [4,5,6,7]. In developed countries, up to 70% of the cereal harvest is used in the daily diet of animals, whereas, in developing countries this commodity is mainly used for human consumption [8]. In addition, plant protein sources, such as by-products from the extraction of oil from oilseed crops, are regularly present in animal feeding and complement the cereal grains which are usually poor in protein [4,6,9,10].

Cereals for the global feed industry include maize, wheat, barley, sorghum, and oats grains [7,9]. Essentially maize, as well as wheat, are considered key global agricultural commodities in regard to farm animal diets [4,11,12,13]. In fact, the majority of the maize production in the world (approximately 55%) goes into animal feed, because maize and products derived thereof are widely used feed raw materials [7,12,13,14,15,16]. Wheat in feedstuffs represents around 20% of the total wheat, with the remainder of the wheat used for human consumption. Nevertheless, in the European Union (EU) almost half of the wheat is used in feed [17,18]. Therefore, wheat grains and the respective by-products are also seen as suppliers of various significant materials in livestock feed [13,18,19]. 

Oilseed crops like soybeans, cottonseed, sunflower, sesame, and palm are also used as vegetable protein sources in the manufacturing of animal feed [9,10]. However, soybean products remain universally accepted as the most important and preferred feed commodities due to their high-quality protein content [4,10,13,20,21,22]. In fact, soybean meal, which is the by-product of oil extraction from soybeans, represents two-thirds of the total world output of protein feedstuffs [20]. 

The global demand for agricultural crops has been increasing over the years, with an expected growth of 84% between 2000 to 2050 [4,11,23,24,25]. This development is intended, in part, to meet the rapid growth and strengthening of the livestock industry, propelled by the rising demand for livestock products [2,10,25]. This is, in turn, driven essentially by increases in world population and urbanization rates, as well as changes in lifestyles and food preferences [10,11,23,25]. Consequently, animal feed safety has become even more of a concern for both producers and governments since feed consumption is, eventually, a potential route for hazards to reach the human food chain [10,25,26,27]. Thus, in accordance with the Directive 2002/32/EC, the quality and safety of products intended for animal feed must be assessed prior to their use in feed to ensure that they do not represent any danger to human health, animal health or the environment, or do not adversely affect livestock production [27,28]. Among the undesirable substances laid down in this Directive, mycotoxins have been increasingly targeted as becoming one of the most important dangers in the raw materials of feed, due to the verified increase in their formation [29,30]. In this review, several constraints posed by these contaminants at the economical and the commercial level will be discussed, along with the legislation established in the European Union to restrict mycotoxins levels in animal feed. In addition, the occurrence of legislated mycotoxins in the raw materials and their by-products of the feeds of interest, as well as in the feed, will be reviewed. Additionally, an overview of the different sample preparation and detection techniques reported for mycotoxin analysis will be discussed.

## 2. Mycotoxins Classes and Toxicity

Mycotoxins are a relatively large and chemically diverse group of toxic secondary metabolites of low molecular weight. They are typically produced by filamentous fungi, especially those belonging to the genus *Aspergillus*, *Penicillium*, *Alternaria*, and *Fusarium*, although *Claviceps* and *Stachybotrys* are also important mycotoxins producers. Approximately 300 to 400 mycotoxins have been identified and reported so far [5,31,32]. However, regarding their prevalence in feeds and their known effects on livestock health, only a few groups of mycotoxins are considered to be a safety and economic concern, namely, aflatoxins (AFs), fumonisins (FMs), ochratoxins (OTs), trichothecenes (TRCs), and zearalenone (ZEN) [5,27,33]. Other mycotoxins, such as patulin, citrinin, and other emerging mycotoxins are beyond the scope of this review. With these relevant classes in mind, a brief introduction about each one will be provided along with the associated toxicological effects.

### 2.1. Aflatoxins

*Aspergillus flavus* and *A. parasiticus* are the main species of aflatoxin-producing fungi, although *A. nomius* and *A. pseudotamarri* are known to produce them, as well. The AFs group encompasses several different toxins, however, only the following four types are most abundant: aflatoxin B_1_ (AFB_1_), B_2_ (AFB_2_), G_1_ (AFG_1_), and G_2_ (AFG_2_) [32,34,35]. The metabolic products derived from AFs are aflatoxin M_1_ (AFM_1_) and M_2_ (AFM_2_) which are also referred to as important contaminants of this class [32,36,37].

AFs represent the group of fungal toxins of greatest concern in terms of human toxicity. Their toxic effects advert from their entry in the human food chain in two ways: (i) First, directly, after human exposure by consumption of contaminated crops or finished processed food products, since aflatoxins are very stable and may resist food processing operations. (ii) Secondly, indirectly from tissues, eggs, milk, and dairy products of animals fed with aflatoxin-contaminated feeds, through excretion of the hydroxylated derivative of AFB_1_ and AFM_1_. Actually, AFB_1_ is the most commonly occurring aflatoxin and most potent hepatocarcinogen, classified by the International Agency for Research on Cancer (IARC) as a human carcinogen (group 1) and AFM_1_ as possibly carcinogenic to humans (group 2B) [33,38,39,40,41,42]. Concerning livestock health, AFs are also a major problem causing acute death to chronic disease. Clinical signs of animal intoxication include gastrointestinal dysfunction, anemia, jaundice, hemorrhage, and an overall decrease in productive parameters, such as reduction in weight gain, lower feed efficiency, decreased egg or milk production, inferior carcass quality, and increased susceptibility to environmental and microbial stressors [32,41,42,43]. Ultimately, prolonged exposure to low dietary levels of AFs can result in extensive functional and structural liver lesions, including cancer. It is important to note that nursing animals, as well, are exposed to the AFB_1_ toxic metabolite secreted in milk [32,41,42,43]. 

### 2.2. Fumonisins

FMs are commonly classified as *Fusarium* toxins since they can be produced by several species of this genus, with *F. verticillioides* (previously classified as *F. moniliforme*) and *F. proliferatum* as the main producing species. However, *A. niger* was recently found to also produce FMs [36,42,44]. Within the 16 fumonisin analogues known to date, the B-series FMs (FBs), which compromise fumonisin B_1_, B_2_, B_3_, and B_4_, are the most important ones [36,42,45].

Fumonisin B_1_ (FB_1_) is reported as the predominant and most toxic member of the FMs family and has been recognized as a possible human carcinogen (group 2B) [38,42,46]. Fumonisin B_2_ (FB_2_) is also toxicologically significant. Apparently, the carcinogenic character of FBs is not related to direct DNA damage, but rather it is associated with the disruption of sphingolipid biosynthesis due to structural similarities of these toxins with the backbone precursors of sphingolipids [36,40,41]. In animals, ingestion of feed contaminated with FBs can cause significant disease in horses, swine, and rabbits which are considerably more sensitive than cattle and poultry [32,41,47]. Leukoencephalomalacia syndrome appears mainly in horses triggering primary symptoms like lethargy, blindness, and decreased feed intake, and ultimately, convulsions and death. In pigs, FB_1_ is associated with pulmonary oedema whose clinical signs typically include reduced feed consumption, dyspnea, weakness, cyanosis, and death [36,40,41]. In addition, these mycotoxins have also shown hepatotoxicity [32,40].

### 2.3. Ochratoxins

Production of the OTs, ochratoxin A (OTA) and ochratoxin B (OTB), occurs essentially by fungi belonging to the genus *Aspergillus* and *Penicillium*, namely by the species *A. ochraceus*, *A. carbonarius*, *P. verrucosum*, and *P. nordicum* [32,36,37,48]. 

OTAs are linked with potent nephrotoxic effects in animals as a consequence of exposure to naturally occurring levels in feed, since the kidneys are the major target organ [32,40,41,46]. In fact, OTAs have been associated with endemic nephropathy in swine [36,46]. High dietary doses of this toxin may cause liver damage and necrosis of intestinal and lymphoid tissue [32,40]. Regarding humans toxicity, OTAs have been implicated in a fatal kidney disease typical in the Balkan countries (Balkan endemic nephropathy) and have been classified as possibly carcinogenic (group 2B) [32,38,41,46]. Additionally, there has been a public health concern with respect to the transfer of OTA to animal-derived food [42].

### 2.4. Trichothecenes

TRCs are produced to a great extent by *Fusarium* species, although not exclusively, since some *Cephalosporium*, *Myrothecium*, *Stachybotrys*, and *Trichoderma* species also produce these mycotoxins. This is a large class of fungal metabolites with more than 150 structurally related compounds, which are chemically divided into four types (A to D) [32,41,43]. TRCs from type A and B are the most important. Type A-TRCs comprises mainly HT-2 and T-2 toxins (HT-2 and T-2), while type B-TRCs are frequently represented by deoxynivalenol (DON), its derivatives 3-acetyldeoxynivalenol (3-AcDON), 15-acetyldeoxynivalenol (15-AcDON) and nivalenol (NIV) [49,50]. 

HT-2 and T-2, although not being very prevalent, are the most toxic members of type A-TRCs [40,41,42,43]. They were found to inhibit protein and DNA synthesis and weaken cellular immune responses, in animals [40,42]. Symptoms include decreased feed intake and weight gain, bloody diarrhea, hemorrhaging, oral lesions, low egg and milk production, abortion, and death in some cases [40,41,42,43].

DON is one of the least acutely toxic TRCs, however, as it is highly incident, it is considered very relevant in animal husbandry [32,40,42,51]. Exposure to DON more severely affects monogastric animals, especially swine, and may cause feed refusal, vomiting, and anorexia, as well as the symptoms described previously for HT-2 and T-2 [32,41,43]. Overall, ingestion of low to moderate levels of this mycotoxin by animals leads to increased susceptibility to pathogens and to a poor performance [32,41]. DON was categorized by IARC as not classifiable with respect to its carcinogenicity to humans (group 3) [38].

### 2.5. Zearalenone

ZEN is a *Fusarium* mycotoxin produced particularly by *F. graminearum* and also by *F. culmorum*, *F. cerealis*, *F. equiseti*, among others, and it has α-Zearalenol (α-ZEL) and β-Zearalenol (β-ZEL) as derivatives [36,41,52]. Once ZEN has structural similarities to the female sex hormone, estradiol, it is classified commonly as a nonsteroidal estrogen. This chemical characteristic gives it the capability of binding to estrogen receptors, causing adverse effects associated with reproductive disorders and hyperestrogenism, both in humans and breeding animals [32,36,37,42]. According to IARC, ZEN belongs to group three, which means it is not classifiable regarding its carcinogenicity to humans [38]. 

## 3. Mycotoxins Economic and Commercial Implications 

Animal consumption of mycotoxin-contaminated crops may cause adverse health effects which include occult conditions (for example, growth retardation, impaired immunity, and decreased disease resistance), chronic to acute disease, and even death. Basically, these hazards affect animal performance to a great extent, representing a global concern for the livestock industry [5,32,46]. Therefore, a threat, such as mycotoxins, to the safety of the feed supply chain becomes a significant constraint to animal production systems [5,53]. These metabolites cause perturbations in the feed industry due to the decrease in the quality of commodities which may even lead to the rejection and disposal of highly contaminated crops [5,32,46]. Naturally, large costs on the economy of these industries arise from mycotoxin contamination. Apart from the aforementioned problems, economic losses may be associated with increased costs for health care, finding alternative feed sources, prevention strategies, investment in testing methods, and for regulations [5,8,32,33]. Additionally, mycotoxins presence may impact on international commodity trade, propelled by increasing globalization [32,34]. 

In an attempt to avoid the adverse effects and implications discussed above, several worldwide institutions and organizations have restricted the accepted levels of certain mycotoxins in animal feeds, since truly mycotoxin-free feedstuffs are impossible to guarantee. Naturally, the limits and the mycotoxins targeted by legislation vary from country to country since different scientific, economic, and political factors influence this decision-making process [26,32,33,43]. 

Particularly, in the EU, the legislations (regulation or recommendation) established so far cover AFs, FBs, OTA, some types of A and B TRCs, and ZEN, in different feeding matrices. Directive 2002/32/EC specifies maximum content for AFB_1_ in products intended for animal feed [28]. Guidance values for DON, FBs, OTA, and ZEN contamination were set in the Commission Recommendation 2006/576/EC [54]. 

## 4. Mycotoxin Occurrence 

Various factors are known to influence the incidence of mycotoxins, despite their unavoidable and unpredictable nature. Their production can start in the field throughout the crop growing cycle and continue during harvesting, drying, processing, and storage steps, strongly depending on various environmental conditions. These comprehend not only climatic factors, such as temperature and moisture content which are the main aspects modulating fungal growth and mycotoxins production, but also pH, bioavailability of micronutrients, and insect damage, for example [32,33,37,46,50,55]. Others factors like geographic location, agricultural practices, harvest year, and the length and conditions of storage affect the extent of the contamination of a particular commodity [32,33,56,57]. However, the substrate susceptibility to fungal invasion plays a major role in mycotoxin production [58]. Moreover, due to the climate changes across the globe, some changes in the distribution and cycles of the molds are expected, since every mold species has its own optimum conditions of temperature and water activity for growth and formation of toxic metabolites. 

In order to understand the mycotoxin prevalence and contamination levels in the main raw materials of feed that are the subject of this work, global mycotoxin occurrence data was gathered in Table A1, Table A2 and Table A3. These tables represent an overview of contamination in maize, wheat, and soybeans and their by-products, respectively, collected by several authors, in the last three years (2016–2018) through searches in PubMED and ScienceDirect. Globally, maize and wheat are by far the most studied matrices, while soybean is the least studied, which is in agreement with previous reports [59]. In all substrates, the raw ingredients themselves were more subjected to mycotoxin contamination surveys than the respective by-products, perhaps because the last ones are usually more complex matrices.

Considering Table A1, it can be pointed out that in 2016 there was an increase in the samples of maize and the derived by-products in which mycotoxins were researched. This may be because this crop is among the most susceptible to mycotoxigenic fungi infection, and also since its production is growing from year to year the need to target these impurities has also raised [12]. The fact that maize by-products are increasingly used in animal diets may also explain the larger number of assayed samples of these feedstuffs [58]. In maize, most studies focused on ZEN and type A-TRCs, followed by the occurrence of AFs and FMs. According to FAO [15,27], maize is especially linked with these two contaminants, having a relevant role in economic losses in maize production [60]. Regarding the levels found, AFB1 was the mycotoxin that exceeded the EU legislative level more often, with a maximum value of 1137.4 µg/kg in a sample of raw-cereal from Kenya [61]. ZEN, T-2, and HT-2 have also been reported to exceed the EU legislative levels in some cases, as reported in Table A1.

Inversely, the wheat samples examined decreased from 2016 to 2018 (Table A2). Concerning the mycotoxins targeted in the reports reviewed, DON was by far the most searched, probably because it is frequently associated with this grain [44]. Nevertheless, ZEN and AFs were also studied in this matrix, and the last one exceeded the EU legislative level in three studies (Table A2).

From Table A3 it is possible to observe the mycotoxins that were studied more and these were AFs, followed by ZEN, and DON. Additionally, fewer samples of soybeans and its by-products were analyzed as compared with maize or wheat, and the by-products were studied more than the raw leguminous. Generally, this substrate is not considered a relevant problem in terms of mycotoxin contamination which may be because of its low moisture content and composition (high protein/carbohydrate ratio) that inhibit fungi growth, and also the better conditions used in the storage of this commodity due to the high price of soybean [37,62]. Nevertheless, the mycotoxin contamination verified in the studies reported was remarkable. In the future, more research on this commodity is needed, especially if this trend of production growth continues, in order to better understand which mycotoxins are most commonly associated with soybean and their by-products and whether contamination levels are of concern.

Overall, it seems that the common association between certain raw materials and a specific mycotoxin contamination profile has led researchers to favor the determination of these same contaminants. However, in addition to the fact that mycotoxins formation is a complex and multifactor phenomenon, worldwide contamination and distribution patterns of fungi and their secondary metabolites are predicted to be affected significantly by climate change scenarios, as a result of the appearance of favorable environmental conditions for fungal proliferation in uncommon places [33,46,53]. Therefore, mycotoxins presence is unpredictable and multi-mycotoxins surveys end up being more realistic and preferred. 

Safety complications arising in the feed manufacturing process include aspects like the practice of mixing different batches of distinct raw ingredients, which creates a new matrix with an entirely new risk profile, and the fact that the majority of mycotoxins are stable compounds that are not destroyed during the storage, milling or high-temperature feed manufacturing process [63]. For these reasons, the knowledge of the occurrence and distribution of mycotoxins in animal feeds is of extreme importance, and it provides the opportunity to determine the direct risk posed to animals. Therefore, occurrence data of these toxins in animal feed, collected by several authors from various countries, from 2016 to 2018 was gathered in Table A4. Globally, AFs, DON, and ZEN were the mycotoxins most studied, but the determination of AFs and ZEN derivatives experienced a great increase, from 2016 to 2018. In this late year, the number of samples analyzed was far less than in 2016. The incidence of the mycotoxin, AFB1, most exceeded the EU legislative level in this kind of samples (Table A4). 

Once formed, most mycotoxins are very stable during harvesting and storage. This draws attention to the need for prevention and control strategies such as hazard analysis and critical control point (HACCP) approaches, good agricultural and manufacturing practices (GAP and GMP), and quality control from the field through to the final product [64,65]. However, contaminated feed may be redirected to less vulnerable animal species, or, ultimately, detoxification methods can be used which involve the addition of feed additives “that can suppress or reduce the absorption, promote the excretion of mycotoxins or modify their mode of action” [30,66,67]. These substances have to be authorized under the feed additive Regulation 1831/2003 amended by Regulation 386/2009 [68]. In this way, the hygienic and nutritional quality of feed is guaranteed, ensuring the safety and productivity of the farm animals [30,65].

It is important to mention that when constructing Table A1, Table A2, Table A3 and Table A4, only papers that quantitatively determined mycotoxins were included and the ones that mentioned explicitly the use of the raw materials for human consumption were excluded. Moreover, year-to-year variations were reduced to the maximum because this parameter is beyond the scope of this review, and whenever the results permitted, the percentage and the average of positive samples was calculated. In addition, since all the information was obtained from different methodologies of analysis with distinct sensitivity and accuracy, the quantitative comparison might be quite complex.

### Co-Occurrence

Natural contamination of raw ingredients and feeds with an array of mycotoxins is a frequent scenario around the world, which can be explained by the ability of molds to simultaneously produce different kinds of mycotoxins and because commodities may be concurrently, or in rapid succession, infected with different fungal species. In addition, composite feed is made up of a mixture of several raw ingredients, making it particularly vulnerable to multiple mycotoxins contamination [5,33]. 

When it comes to the reports considered for this review, several described this phenomenon within the regulated mycotoxins. In maize and products derived thereof, Chen et al. [69] found co-contamination with AFB_1_ and ZEN. Chilaka et al. [70] reported that 60% of maize samples were infected with at least two mycotoxins and FBs co-existing with DON in 11% of samples. ZEN and DON were simultaneously found by Dagnac et al. [71], who reported a frequent co-contamination of more than one mycotoxins in the samples under analysis. Jovaišienė et al. [72] found DON and ZEN co-occurring in all samples of silage and DON, ZEN, and T-2/ HT-2 co-occurred in all fermented silage samples. Kamala et al. [73] detected that 33% of the samples were contaminated with both AFs and FB_1_ + FB_2_. Kosicki et al. [15] frequently identified this phenomenon in their study, with the combination of DON and ZEN being the most prevalent, however, the co-occurrence of DON, T-2 and HT-2; ZEN, T-2 and HT-2; DON, T-2, HT-2, and ZEN; and DON and FMs were commonly found in maize. While in maize silage, apart from these groups, the co-existence of DON and OTA; DON, OTA and ZEN; ZEN and OTA; and T-2, HT-2, and OTA were also detected. Mngqawa et al. [74] reported the occurrence of a wide variety of mycotoxins in their samples with relevance to AFs and FBs. Finally, Murugesan [75] verified that 50% of samples were contaminated with more than one these analytes. In wheat, co-occurrence between ZEN and DON was found by Calori-Domingues et al. [76], in several samples. Already in soybeans and derived by-products, Egbuta et al. [77] showed that there was simultaneous occurrence of AFs and FB_1_. Regarding animal feed, Hu et al. [78] concluded that combinations of two mycotoxins were more frequent than three but highlighted the presence of AFB_1_, OTA, and ZEN. Kongkapan et al. [79] detected that AFs co-occurred with ZEN and AFB1 with DON. Kosicki et al. [15] frequently identified this phenomenon with combinations of DON and ZEN; DON, T-2 and HT-2; ZEN, T-2 and HT-2; DON, T-2, HT-2, and ZEN; DON and FMs; DON and OTA; DON, OTA, and ZEN; ZEN and OTA; and also T-2, HT-2 and OTA. Lastly, a high incidence of co-contamination was reported by Kovalsky et al. [43]. Globally, these results are in line with the statements that combinations of two mycotoxins occur more frequently [32,33]. It was verified that DON and ZEN along with AFB_1_ and FBs were commonly reported as co-existing in the reviewed samples, followed by DON and FBs as well as DON, ZEN, and HT-2/T-2.

Multiple mycotoxin contaminations pose great concerns since it is completely stated that adverse effects on animal health and performance can be additive and/or synergistic, which means that the overall toxicity is not only the sum but the multiple of the individual toxicities of the mycotoxins [5,80]. This means that the study of just one of these toxins provides insufficient information about the risk associated with a respective feedstuff and that attention toward mycotoxin co-occurrence should be increased [42,81]. Additionally, the use of raw materials from different types and origin contributes to increase the likelihood of multi-mycotoxin contamination, including nonregulated compounds, usually called “emerging mycotoxins”. The presence of conjugated mycotoxins, sometimes in amounts similar or even higher than the corresponding free mycotoxins, is another issue that deserves detailed attention, although it is out of the scope of this review. However, the potential for biological effects remains, and the toxicological potential can be substantial enhanced.

Currently, legislation over the world, including in Europe, only considers mycotoxin mono-exposure data and does not address relevant mycotoxin combinations, which is considered a loophole that should be taken into account in the future. 

## 5. Mycotoxin Determination Methods 

Evaluation of mycotoxin contamination on feed materials and feed is a direct requirement of the adoption of legislation limits for these compounds, providing information to producers, manufacturers, traders, and researchers [43,63,82,83]. Moreover, analytical data are fundamental for assessment of the potential risk to livestock and for global trade of their commodities, in the diagnosis of mycotoxicosis, and in monitoring mitigation strategies [84,85]. The determination of these contaminants is quite complex since they represent structurally diverse chemical groups which frequently appear in low concentrations, in a vast range of matrices, and sometimes in various combinations [56,57,85]. Nevertheless, sufficiently reliable, accurate, sensitive and selective methods are available for the qualitative and quantitative analysis of these secondary metabolites. As previously mentioned, feed may also contain the so-called “emerging mycotoxins” and/or conjugate mycotoxins, however, the analytical methods used for these are beyond the scope of the present review. Generally, the following three steps are involved in testing for mycotoxins: sampling, sample preparation, and analytical procedure. 

### 5.1. Sampling

Obtaining sufficiently representative samples of a batch, in other words sampling, is crucial in the entire process of mycotoxins determination. In fact, this step accounts for the greatest source of error since the analytes under discussion often appear unevenly distributed and in trace levels [82,86]. Thus, sampling plans for different commodities were established by several agencies. In the EU, Regulation No 691/2013 amending Regulation No 152/2009 describes methods of sampling in feed materials for the official control of AFs and other mycotoxins. Briefly, representative laboratory samples are prepared from the sampling points by (i) selecting one or more characteristic lots, (ii) repeatedly collecting incremental samples at numerous single positions in the lot, (iii) forming an aggregate sample by combining the incremental samples by mixing, and (iv) preparing the final samples by representative dividing [87,88]. 

### 5.2. Sample Preparation

Sample preparation steps, grinding and subsampling, accomplish the conversion of the aggregate sample into a representative subsample, from which is prepared the laboratory sample. Ideally, a subsampling mill is used, performing these two processes simultaneously. However, a conventional grinder can also be used, where the aggregate sample is crushed, and then a subsample is taken. In the Annex II of the Regulation No 401/2006, it is possible to withdraw the criteria for sample preparation, although it is for the official control of the levels of mycotoxins in foodstuffs [89].

### 5.3. Analytical Procedure

For the majority of the methods, the analytical procedure includes a step of sample pretreatment where mycotoxins are (i) solvent-extracted from the laboratory sample, (ii) the obtained extract is further purified to remove unwanted co-extracted matrix components, and finally (iii) an optional sample concentration step takes place, before the final separation and detection steps.

The following sections provide an overview of the different sample pretreatment techniques and detection methods that have been reported for mycotoxin analyses in maize, wheat, soybeans, their by-products, and animal feed, published in the last years. Additionally, included are enzyme-linked immunosorbent assay as well as gas and liquid chromatography methods that are applied in this field of analysis.

#### 5.3.1. Sample Pretreatment

Sample pretreatment makes it possible to obtain an enriched extract of the compounds of interest, as clean as possible, reducing matrix effects. As there is a great diversity in these techniques, a careful choice has to be made depending on the type of matrix, the physicochemical properties of the target analyte(s), and the final detection method.

##### Extraction with Solvents—Classical Solid-Liquid Extraction

In solid-liquid extraction (SLE), a mixture of solvents or a solvent is intended to extract the analyte quantitatively from the solid sample, with as little additional compounds as possible [82]. As the majority of the mycotoxins are soluble in polar and slightly polar solvents and insoluble in apolar solvents, mixtures of organic solvents, such as acetone, acetonitrile (MeCN), chloroform, dichloromethane, ethyl acetate, and methanol (MeOH) are often used. Small amounts of diluted acids (i.e., formic acid, acetic acid, and citric acid) or water are usually added to improve the extraction efficiency, since an acidic environment can break interactions between the toxins and other sample constituents like proteins or sugars, and water increases penetration of the solvent into the material [57]. Following the addition of the extraction solvent, shaking is used to favor the procedure, and then centrifugation or filtration is normally carried out, before concentration and/or cleanup steps [57,82]. Since the selection of a suitable extraction solvent is a challenging process during the optimization of a method, it is common to test different extraction mixtures in order to understand which one is capable of yielding the highest recovery rates [90]. For example, Sifou et al. [90] tried MeCN/water/formic acid (89/10/1 *v*/*v*/*v*), MeOH/water/formic acid (89/10/1 *v*/*v*/*v*), water/MeCN (84/16 *v*/*v*), MeCN/water/acetic acid (79/20/1 *v*/*v*/*v*), MeOH (100%), and MeCN (100%) to extract OTA in poultry feed samples, concluding that MeOH (100%) provided the most efficient extraction.

##### Instrumental Solvent Extraction—Microwave-Assisted Extraction

Microwave-assisted extraction (MAE) is a relatively quick process that through highly localized temperature and pressure causes selective migration of target compounds from the material to the surroundings using microwave energy [57,91]. A pretreatment technique using MAE followed by solid-phase extraction (SPE) was successfully developed by Chen and Zhang [91] to determine AFs in grains and grain products with liquid chromatography (LC) coupled to a fluorescence detector (FLD). To perform MAE, 12 mL of MeCN were added to 3 g of sample. This mixture was then heated at 80 °C for 15 min and 350 psi.

##### Instrumental Solvent Extraction—Ultrasonic Extraction

Ultrasonic extraction (USE) uses acoustic cavitation to cause molecular movement of the solvent and sample, aggressively improving the transfer of the analytes from the matrix into the solvent with improved efficiency. This technique is carried out in an ultrasonic bath and the duration of the ultrasound application depends on the matrix [82]. Generally, USE enables the reduction of the extraction time, consumes low solvent, is economical, and offers a high level of automation as compared with traditional extraction methods [82,92]. For example, Fan et al. [93] ultra-sonicated the sample together with MeCN 50% for 40 min at 40 °C in order to quantify DON and its derivatives in feed with an ultra-high-pressure liquid chromatography (UHPLC) coupled to the MS/MS method.

##### Cleanup Methods—Solid-Phase Extraction

Solid-phase extraction (SPE) is a technique commonly applied to solid matrices as a purification and/or concentration step, after the extraction of mycotoxins [57]. For the analysis of FB_1_ in soya bean meal and feed and T-2 in corn, Abdou et al. [63] developed a high performance liquid chromatography (HPLC) coupled to FLD (HPLC-FLD) in which the cleanup was performed using a Sep-Pak C18 column eluted with 15 mL of MeOH/water (60/40 *v*/*v*). In an LC coupled to tandem MS (MS/MS) method (LC-MS/MS), this C18 reverse-phase SPE column was only used by Chilaka et al. [70] to determine FBs, DON and 15-AcDON, ZEN and its metabolites, and HT-2 in maize. Relatively to SAX columns, they were merely employed to purify FBs and further detect them with HPLC-FLD, in soya bean seeds and processed soya bean powder [77] and in maize [73,94]. Plus, for example, grade polypropylene depth hydrophilic-lipophilic balance (GPD HLB) SPE column was applied in UHPLC-MS/MS to determine DON and its derivatives in feed after the extraction with MeCN 50% [93].

##### Enhanced Solid-Phase Extraction—Immuno-Affinity Columns

Immuno-affinity columns (IACs) are a particular case of SPE, based on the principle of antigen-antibody interactions [82,87]. IACs allow a highly selective purification, resulting in cleaner extracts with minimal interfering matrix components and low LOQ [82,95]. Although this is an automated sample cleanup method, it is time and solvent consuming, requires a high level of expertise, and the use of expensive disposable cartridges [82]. Moreover, in the presence of low concentrations of organic solvents, the denaturation of the antibodies is verified, which means that the extract must be an aqueous solution containing little or no organic solvent. Besides, there is the possibility of nonspecific interactions occurring due to cross-reactivity with other mycotoxins [57]. Differently, in multi-mycotoxins LC-MS/MS surveys, multiple IACs that allowed the specific capture of multiple mycotoxins were just used by Hu et al. [78] in feed, and Zhang et al. [96] in corn and wheat.

##### Enhanced Solid-Phase Extraction—Multifunctional Columns

Multifunctional columns (MFCs) allow the performance of a one-step purification process where compounds, such as proteins, fats, pigments, etc., that may interfere in the analytical method are retained in the solid phase, allowing the analytes of interest to pass through the column, at the same time [57,82,95]. The MycoSep^®^/MultiSep^®^ columns, suitable for mycotoxins, are filled by adsorbents such as charcoal, celite, ion-exchange resins, polymers, and other materials, packed into a plastic tube between two filter discs. Overall this is a simple and quick process because it does not require the washing and elution steps [57,82]. Plus, MFCs eliminate the problems of irreversible adsorption or premature elution of analytes from the sorbent material [82]. In raw feed ingredients and feed analysis for mycotoxin contamination, MycoSep^®^ 226 and 227 and MultiSep^®^ 211 were the MFCs most used. For example, Wu et al. [97] applied MycoSep^®^ 226 column to clean extracts for the subsequent determination of AFB_1_ in corn and by-products, wheat and by-products, soybean meal, and diverse feeds with HPLC-FLD. The MycoSep^®^ 227 column was used for TRCs analysis in wheat with a GC-MS method [98]. Finally, Kosicki et al. [15] reported the employment of the MultiSep^®^ 211 column to purify maize and feed extracts to further quantify FBs with LC-MS/MS. Additionally, the MycoSep^®^ 224 and MycoSep^®^ 225 columns were used for the determination of ZEN and DON, respectively, in wheat with HPLC coupled to diode array detection (DAD) (HPLC-DAD) [76].

##### Enhanced Solid-Phase Extraction—Molecularly Imprinted Polymers

Molecularly imprinted polymers (MIPs) represent a purification method based on the chemical creation of simulated binding sites using a template molecule for the analytes of interest, in a cross-linked polymer matrix. The target molecule is retained as a result of the shape recognition [57,82,87].

This technique has some potential given its high selectivity and great stability to heating and pH shifts, as well as being considered a cheaper alternative for cleanup [57,82]. However, their development and optimization require considerable time, which includes finding the best template molecule for imprinting and testing the resultant material in relevant applications [99]. Additionally, MIP are applied usually for determination of one analyte. Wang et al. [100] developed a solid-state electrochemiluminescence sensor that combined with the MIP technique allowed ultrasensitive determination of OTA. This sensor was successful applied to OTA determination in real corn samples, obtaining recoveries ranging from 88.0% and 107.9%.

##### Combined Extraction/Clean-up/Concentration—QuEChERS

The QuEChERS method, which means quick, easy, cheap, effective, rugged, and safe, even though it was not initially developed for the analysis of mycotoxins, has been successfully applied with this objective [87,101]. It involves a micro-scale extraction using MeCN, followed by a salting-out step of the analytes into the MeCN phase and then a purification based on a quick dispersive SPE. Basically, in the extraction step, MgSO_4_ and NaCl are used to reduce sample water, while in the purification step simple sorbent materials like primary secondary amine (PSA), C18, and alumina are used to retain co-extracted compounds [57,87,101]. With the aim of ensuring an efficient extraction of mycotoxins, the original method may suffer some modifications, for example, changes in the salts used, in their quantity or in the amount of C18, or addition of formic acid, water or MeOH to the extraction solvent. Plus, in dried matrices, a swelling step with water is recommended to make samples more accessible to the extraction solvent [57]. Xu et al. [102] applied a modified QuEChERS procedure to extract DON and its derivatives from wheat. The extraction was performed with water, MeCN, and salts (MgSO_4_ and NaCl), followed by the use of *n*-hexane to remove fat. An Oasis^®^ MAX SPE cartridge was used to clean up the extract before the injection in the UHPLC-DAD system. This method allowed good recoveries to be obtained, between 80.0% and 102.2%. Bryla et al. [103] prepared wheat samples for multi-mycotoxins determination with UHPLC combined to high-resolution MS (HRMS), applying a modified QuEChERS procedure. The extraction solvent consisted of a mixture of water and 10% formic acid in MeCN, to which MgSO_4_, NaCl, sodium citrate dihydrate, and sodium citrate dibasic sesquihydrate were added. Then, to eliminate the lipid faction, hexane was used. Finally, MgSO_4_, C18 silica gel, neutral alumina, and PSA were added to perform cleanup. With [104], which aimed multi-mycotoxins analysis in feed, a QuEChERS-based approach performed in one step was chosen. So, water along with MeCN containing 1% acetic acid and MgSO_4_, NaCl, sodium citrate, and disodium citrate sesquihydrate were used. The extract was then analyzed using a UHPLC-HRMS system. 

##### Combined Extraction/Clean-up/Concentration—Matrix Solid-Phase Dispersion

Matrix solid-phase dispersion (MSPD) consists of mixing a small amount of sample with an abrasive solid support material that has been derivatized to produce a bound organic phase on its surface (SPE sorbent), using a mortar and a pestle. According to Ye et al. [105], this technique was extensively applied to solid and semisolid samples for the extraction of drugs, pesticides, pollutants, among others. However, in mycotoxins quantification, MSPD is an unconventional alternative for classical SPE. In the field of analysis reviewed here, Ye et al. [105] developed a new simple and efficient MSPD procedure coupled to HPLC-DAD for the determination of FB_1_ and FB_2_ in corn. Various conditions were optimized, namely the type, volume, and pH of the eluting solvent, the dispersion sorbent, and the ratio of dispersing material to the matrix. They concluded that 10 mL of MeOH with 10 mM formic acid was the eluting solvent that provided better recoveries, with a C18 sorbent in a 2:1 ratio of sample:sorbent.

##### Combined Extraction/Clean-up/Concentration—Dispersive Liquid-Liquid Micro-Extraction

Dispersive liquid-liquid micro-extraction (DLLME) is a novel miniaturized extraction technique in which there is a rapid injection of a mixture of extraction solvent (organic) and dispersive solvent (water-organic miscible) into an aqueous solution that contains the analytes. This leads to the formation of a cloudy solution, and consequently the very large surface area formed between the two phases, and the analytes are enriched rapidly and efficiently in the extraction solvent. After centrifugation, they can be separated in the sedimented phase [57,82]. Although DLLME is more appropriate for aqueous samples, it is possible to apply this method to solid samples after an adequate pretreatment [57]. A novel, rapid and efficient two-step micro-extraction technique, based on the combination of ionic-liquid-based DLLME (IL-DLLME) with magnetic SPE, was developed by Zhao [106], for the preconcentration and separation of AFs in animal feedstuffs before HPLC-FLD. The ionic liquid extractant, 1-octyl-3-methylimidazolium hexafluorophosphate, was used in DLLME to extract AFs in the sample extracting solution medium. Then, hydrophobic pelargonic acid modified Fe_3_O_4_ magnetic nanoparticles were used as an efficient adsorbent to retrieve the AFs-containing ionic liquid from the DLLME step. Therefore, the target of the magnetic SPE was the ionic liquid instead of the mycotoxins. The authors compared the proposed method with other HPLC-FLD in which the cleanup was done with IAC and found no significant differences between data obtained by the two methods at a 95% confidence level.

#### 5.3.2. Detection

A broad range of techniques can be used for this purpose and are generally divided into two categories which are screening methods and chromatographic methods coupled to different detectors. Currently, EU regulations do not require specific methods for the determination of mycotoxin levels, but any method of analysis should be characterized by the criteria defined in Annex III of the Regulation No 882/2004 [107]. Additionally, and although it is for the official control of the levels of mycotoxins in foodstuffs, Regulation No 401/2006 amended by Regulation No 519/2014 lays down, in the Annex II, the specific requirements that the method shall comply with in relation to individual mycotoxins [89,108].

##### Screening Methods

Usually, screening assays are developed in the form of kits which are extremely relevant tools for monitoring mycotoxin in feed ingredients and feed either by analysts with time constraints for making decisions or by those where other methods may not be available due to cost or situation [57,99]. These methods for single or whole mycotoxin classes compromise both qualitative tests that show the presence or absence of the target impurity and tests that yield semi-quantitative or quantitative results [57,109]. Immunoassay-based methods, biosensors, and non-invasive techniques are among the screening methods.

##### Immunoassay-Based Methods

Methods based on immunoassays are settled in the recognition of specific antibodies with mycotoxins that act like antigens [57,109]. Detection is typically facilitated by the presence of a marker. This compound can be radioactive, chromogenic or fluorescent and reacts with an enzyme, generally horseradish peroxidase (HRP). Immunoassays without the marker are based on the natural fluorescence of some mycotoxins, or in measures of conductivity [57]. These tests are preferably employed for the first level screening and survey studies on mycotoxin contamination due to their simplicity, cheapness, sensitivity, and selectivity, although cross-reactivity with structural analogues can occur [57,110]. Plus, they do not require sophisticated equipment or skilled personnel [109]. However, the signal obtained from these techniques can be influenced by co-extractives and by nonspecific interactions or matrix effects [99]. Additionally, in the new scenario of mycotoxin investigation, immunoassay-based methods may have a potential limitation related to the overall selectivity for only one mycotoxin or a small group of compounds, making difficult the simultaneous determination of different compounds and the detection of unknown toxins and conjugated mycotoxins [57,110]. Nevertheless, these methods are in continuous development in various formats, aiming to provide rapid, portable and easy to operate systems [110]. Enzyme-linked immunosorbent assay (ELISA), lateral flow immunoassay (LFIA). and fluorescence polarization immunoassay (FPIA) are included in this category of screening methods [57,99].

##### Enzyme-Linked Immunosorbent Assay

Enzyme-linked immunosorbent assay (ELISA) methods represent a commonly used immunoassay method to rapidly monitor mycotoxins and are routinely used by agro-food laboratories [57,82,101]. For all regulated mycotoxins there are commercially available ELISA microtiter plate kits that have well-defined applicability, analytical range, and validation criteria [82,109,110]. There are several ELISA formats commonly accessible, however, in this field of analysis the predominant form is the competitive one. This is a strategy normally used when the antigen is small and has only one antibody binding site (epitope), which is the case of mycotoxins [82,111,112]. The competitive format is characterized by the fact that the signal intensity is inversely correlated with the concentration of antigen in the sample [113,114]. Within this format type, it is possible to distinguish the classical competitive ELISA and the competitive inhibition ELISA [113]. The classical competitive format consists of the immobilization of the antigen standard on the surface of the plate. Then, there is an incubation of the antibodies directed against the target mycotoxin with the sample. The antigens in the sample will compete with the immobilized ones for binding to these antibodies. After the washing step, the antibodies bounded to the analyte are rinsed away [113]. In this case, detection can be performed directly or indirectly, which mainly determines the sensitivity of an ELISA. Direct detection uses an enzyme-labelled primary antibody that reacts with the antigen, while an enzyme-labelled secondary antibody with affinity for the primary antibody is used in indirect detection [111]. In the competitive inhibition format, the competition occurs between unlabeled antigens from samples and enzyme-labelled antigens (enzyme conjugate) for binding to an antibody directed against the target mycotoxin. In this format, the plate can be coated with capture antibodies with affinity for the analyte or for a primary antibody [111,113]. Common to both types of competitive assays is the addition of an adequate substrate that is allowed to incubate so that the enzyme that conjugated with the antibody or antigen (classical or inhibition format, respectively) can act and produce changes in a given parameter [111,112,114]. A large variety of substrates are available, and the choice depends upon the required assay sensitivity and the instrumentation available for signal-detection, although a mixture of hydrogen peroxide and a chromogen are usually applied [111,112]. Indeed, the simplest detection is a visual color change which provides qualitative and semi-quantitative results [57]. The last step of all assays is the addition of a stop solution causing the reaction between the enzyme and the substrate to stop. The results are usually determined in a plate reader. The signal intensity weakens as the sample antigen concentration increases, since a larger quantity of analyte results in either fewer enzyme-labelled antibodies bound to the antigen adsorbed to the plate (classical format), or less enzyme-labelled antigens bound to the antibody on the plate [112,113,114]. Advantages of ELISA include, in addition to the specificity of antibody-antigen binding, a relatively low limit of detection (LOD), high sample throughput with low sample volume, minimal cleanup procedures, and ease of application [82,109]. However, this method is not so reliable in the case of complex matrices, since it is quite time-consuming and the kits are for single use and are not suitable for field-testing [57,82,87,109]. In addition, the possibility of false positive and false negative results requires additional confirmatory analysis [82,109]. From Table A5, where the ELISA methods are reviewed, it is possible to conclude that all analytes were quantified with competitive ELISA after SLE mainly with an aqueous solution of MeOH or with water. Additionally, absorbance was the detection method most used, followed by optical density (OD), while FLD was only used by [115] to detect OTA in corn. Regarding mycotoxins studied with ELISA, the more targeted mycotoxins were AFs and DON.

##### Lateral Flow Immunoassay

Lateral flow immunoassay (LFIA) or membrane-based test strips are commercially available in the form of kits providing mainly visual qualitative results that indicate the presence or absence of a specific mycotoxin below a predetermined fixed level [57,116]. More recently, semi-quantitative detection is possible using a portable photometric strip reader [99]. In LFIA, the sample flows along the strip by capillary migration and two lines are formed, the test line whose intensity is inversely correlated to the mycotoxin concentration, and the control line that allows the assay validation [57,109]. This is an inexpensive screening tool that enables rapid, one-step, and in situ analysis [57,82,109]. Nonetheless, LFIAs often show false-positive results due to matrix interferences and reproducibility and sensitivity problems [57,109]. Chen et al. [69] developed and optimized a multiplex LFIA for the simultaneous on-site determination of AFB_1_, ZEN, and OTA in corn. This device provided both qualitative and quantitative results. LFIA was also used by Carvalho et al. [117] to evaluate mycotoxin presence in corn silages. FM, DON, AF, OTA, ZEN, and T-2/HT-2 were quantified using Reveal Q+ kits from Neogen Corporation. Beloglazova et al. (2017) developed a flow-through membrane–based assay for the screening of four mycotoxins DON, ZEN, OTA, and AFB_1_ in feed matrices. This approach allowed the separation of different test zones, and therefore minimized the across-influence. 

##### Fluorescence Polarization Immunoassay

Fluorescence polarization immunoassay (FPIA) indirectly measures the rate of rotation of a fluorophore (tracer) in solution based on the competition between the free mycotoxin on the sample and the mycotoxin labelled with the tracer towards a specific antibody. When tracers bind to the antibodies their rotation is restricted, and consequently, the fluorescence polarization value increases. Therefore, if a sample has a high concentration in the target mycotoxin it competes with the tracer for the interaction with the antibody resulting in free tracers with a faster motion, in other words, a low fluorescence polarization signal. Basically, this value is inversely proportional to the amount of free mycotoxin in the sample. The FPIA is reliable, rapid, easy to perform, and relatively suitable for automation, however, their solution-based nature makes it less easy to use in field scenarios [57,109,118]. Concerning mycotoxin analysis in raw feed ingredients and feed, Li et al. [119] developed a homologous and high-throughput multi-wavelength FPIA for the multiplexed detection of DON, T-2, and FB1 in maize flour with an LOD of 242.0 µg/kg, 17.8 µg/kg, and 331.5 µg/kg, respectively.

##### Biosensors and Biosensor-Based Methods

Biosensors or immuno-sensors are analytical devices composed of one antibody which is a recognition element that reacts in a sensitive and selective way towards the target mycotoxin, and a transducing element which is responsible for converting the change of the physical variable produced by the reaction into a measurable signal [57,109]. In fact, antibodies are the most widely used recognition element in sensors but there is an extensive range of other components [87,120]. Alternatives to this classic element include, among others, enzymes, peptides, aptamers, and MIPs [87]. Similarly, techniques comprised of various transducing elements are available and are commonly applied with an optical or electrochemical nature, along with the piezoelectric and magnetic systems [120]. Optical detectors can be based on surface plasmon resonance, fluorescence, optical waveguide light mode spectroscopy, and total internal reflection ellipsometry. Electrochemical detectors are based on potentiometry with a carbon working electrode, differential pulse voltammetry, conductometry, etc. [57]. These methods are very promising since they provide results in a faster way, have a low price, high-throughput, greater sensitivity, and are portable [57,87,109]. However, they rely on specialist analytical equipment and their low selectivity and reproducibility make it necessary to confirm the results [57,87]. Plus, their applicability to routine analysis needs to be further investigated. Several authors developed biosensors and biosensors-based methods for mycotoxin analysis in raw feed ingredients and feed. For example, electrochemical immunosensors were designed to determine AFB_1_ in maize [121,122] and for FB_1_ and DON determination in the same matrix [123]. Plotan et al. [124] applied an innovatively biochip array technology to multi-mycotoxin semi-quantitative screening in a large variety of feed ingredients, obtaining an overall average recovery of 104%. An optical aptasensor was developed based on the hybridization chain reaction amplification strategy and fluorescent perylene probe/DNA composites for ultrasensitive detection of OTA [125]. The application to corn samples demonstrated the feasibility and potential of the proposed enzyme-free amplification method in the practical applications of agricultural products. Wang et al. [126] developed a novel and ultrasensitive aptamer-based biosensor for the detection of AFB_1_ in corn. For this, fluorescent nitrogen-doped carbon dots were synthesized and assembled on aptamer-modified gold nanoparticles.

##### Noninvasive Methods

Some noninvasive methods have been developed to assess mycotoxin contamination allowing simple, rapid, and in situ analysis. These kinds of methods enable decisions to be made promptly and avoid possible loss of an entire lot. However, due to the high matrix dependency and lack of appropriate calibration materials, their application is still limited. The nondestructive approaches include infrared spectroscopy (IR) techniques and Raman spectroscopy [57,82].

##### Infrared Spectroscopy

Promising IR techniques include near-infrared (NIR) spectroscopy either in combination or not with Fourier-transform (FT-NIR). Basically, NIR spectroscopy is based on the measurement of the absorption or reflectance of a given incident NIR radiation in the sample. The exposition to radiation in this region of the spectrum causes a change in the energy of the chemical bonds involving hydrogen (for example, C-H, N-H, O-H, and S-H). However, the bands observed in the NIR spectral region are very difficult to assign to specific compounds because of the complexity of the samples and also due to spectra overlapping and interference from other functional chemical groups. This implies the application of modern chemometrics methods in the calibration development process. The detection of the NIR radiation absorbed by the sample is conducted by transmittance, reflectance, interaction, and/or transflectance measurement [57,82]. This promising technique requires minimal or no sample pretreatment and it is environmentally friendly, and therefore it does not require reagents and does not produce chemical waste [82,127]. In addition, NIR is highly accurate, needs little expert training, and has the ability to analyze both large and small quantities of feeds which avoids errors associated with inconsistent sampling [128]. Beyond the difficulties in the interpretation of spectral data posed by this technique, other drawbacks are related to the fact that NIR is only useful at high contamination levels, as well as the system is heavily dependent on the establishment of an accurate calibration procedure [57,128,129]. A nondestructive detection of DON by ultraviolet-visible near-infrared diffuse reflection spectroscopy in unprocessed, solid maize kernels was investigated by Smeesters et al. [130]. They proposed a two-stage measurement methodology enabling efficient monitoring of the local DON-contamination on a large number of maize kernels. Mignani et al. [131] presented a novel chemometric classification for FTIR spectra of mycotoxin-contaminated maize at regulatory limits. They investigated the classification ability of a decision tree at 1750 µg/kg for DON in maize, which corresponds to the regulatory limit set by the EU for unprocessed maize in food.

##### Raman Spectroscopy

The principle behind Raman spectroscopy relies on the irradiation of matter with monochromatic light to further detect the loss of energy in the form of scattered light. Thus, information about the vibrational transition energy of the molecules is provided by this technique. Symmetrical vibrations of the covalent bonds in nonpolar groups (e.g., C = C) enhance the sensitivity of Raman spectroscopy [129,132,133]. This method provides a unique expression of the molecular structure, and therefore it is considered to be a molecular fingerprint providing more useful qualitative and quantitative information on chemical functional groups of mycotoxin compounds and its derivatives than the conventional spectroscopic techniques [132,133]. Despite this advantages, Raman spectroscopy has received remarkably little attention for detection of mycotoxins in grains and oilseed [133]. In 2016, Lee and Herrman [134] investigated the potential and feasibility of a surface-enhanced Raman spectroscopy (SERS) method as an alternative accelerated technique to screen ground maize contaminated with FMs. Chemometric models developed based on SERS spectra showed an acceptable predictive performance and ability for qualitative and quantitative analysis.

##### Chromatographic Methods

Chromatographic separation coupled to a suitable detection system is the most widely used strategy to quantitatively analyze mycotoxin contamination, unambiguously confirm positive findings, and also serve as a reference method to validate other tests. These are methods which are highly selective, accurate, and reproducible that need expensive instrumentation and chromatographic expertise. In feed analysis, LC is the most common method, although gas chromatography (GC) and thin layer chromatography (TLC) are still considered [82,109,110].

##### Thin Layer Chromatography

Contrary to what happens in developed countries, TLC is a method that is still commonly used in countries under development, especially if coupled to an ultraviolet (UV) or fluorescence scanner [82,99]. TLC allows qualitative and semi-quantitative determination of naturally fluorescent mycotoxins. The qualitative confirmation can be done through the retention factor value and the fluorescence color after comparison with an external standard. In semi-quantification, the sample is compared with authentic standards using the visual estimation of fluorescence of the separated spots under long wavelength UV light. Therefore, with this approach precise and reliability results depend directly on skilled and experienced people. Quantification is mainly achieved by measuring fluorescence intensity or absorbance when separated spots on the TLC plate are exposed to UV light. TLC can be applied both in one- and two-dimensional formats. This method makes possible rapid analyze of several samples in a short period of time, has a low cost per sample analyzed, and easy estimation of contamination levels [82]. However, low sensitivity and reproducibility along with the need of large quantities of solvent, intensive laboratory procedures. and difficulties in automation have led TLC to be commonly replaced by other chromatographic techniques [82,87]. Betancourt and Denise [135] applied this method to screen AFs contamination in corn hybrids. The TLC plates were exposed to UV light at a short wavelength (250 nm) and visual comparison to standards allowed the identification of positive samples. Mona et al. [136] performed AFB1 detection in cattle feed with TLC, where standard and test samples were inspected under a long-wave UV lamp (360 nm).

##### Gas Chromatography

In GC, volatile compounds are separated into open tubular columns coupled to a variety of general or specific detectors. GC coupled with an MS detector (GC-MS) simultaneously allows the identification and quantification of compounds, and based on these reasons is the first choice in mycotoxin analysis [57,137]. The methods of GC-MS are described mainly for TRCs and mainly in wheat, generally after extraction of the compounds with MeCN, cleanup with MFCs (Table A6) and derivatization [57,87,109]. The derivatization procedure aims to counteract the low volatility and the high polarity of many mycotoxins, and therefore allow their analysis. The silylation and acylation reactions are the most common approaches, converting mycotoxins into more volatile, less polar, and thermally more stable derivatives. In silylation, the introduction of a silyl group by a silyl reagent is valuable for the MS applications because it produces either more interesting diagnostic fragments or ions with particular characteristic ions for single ion monitoring (SIM). The derivatization method is applied majorly when detecting mycotoxins with GC-MS. Alternatively, acylation is preferable when acylated compounds are more stable than the silylated compounds [57,137]. (Table A6). The GC-MS methods allow for the reliable and sensitivity determination of multi-mycotoxins in one single run. 

##### Liquid Chromatography

Liquid chromatographic methods are the mainstay separation method for mycotoxin analysis. Several variations of LC are available offering good sensitivity, high dynamic range, and versatility. On the other hand, these methods suffer from portability, cost, and issues related to the sample type such as the matrix effect, the choice of calibration, and the sample preparation [82,87].

HPLC is a well-established and prevalent method for the identification and quantification of mycotoxins [109]. To date, both normal- and reverse-phase columns have been used for this purpose. However, the great majority of separations are performed on reverse-phase columns because the majority of mycotoxins are soluble in polar organic solvents such as methanol, acetonitrile, water, and in their mixtures. This HPLC procedure relies mostly on C18 columns and mobile phases composed of mixtures of water with MeOH and/or MeCN in proper ratios [82,99]. HPLC has high separation power, is easy to use, and suitable for automation [82]. Traditionally, this chromatographic method is equipped with spectrometric detectors like UV (HPLC-UV) and fluorescence that depend on the analyte. From Table A7, it is possible to see that HPLC-UV was not used only once. The studies [97,138,139,140] applied this technique to quantify DON, ZEN, and OTA in raw feed ingredients and feed. On the contrary, FLD was abundantly used, after SLE mostly with MeOH and cleanup by IACs, to analyze mainly AFs, and also FBs, T-2, ZEN, and OTA in those matrices. Commonly, pre- or postcolumn derivatization procedures are used to improve mycotoxins fluorescence properties, and consequently increase sensitivity. In the precolumn approach, trifluoroacetic acid is majorly applied, converting AFs into their corresponding hemiacetals derivatives which have stronger fluorescence. However, since this is a toxic and corrosive chemical and the derivatives formed have relative instabilities, this is not the preferred method. Additionally, postcolumn derivatization offers the advantage of being automated [82]. Therefore, this strategy is applied more to detect mycotoxins (Table A7). Different methods can be used, such as bromination by an electrochemical cell (Kobra cell) which is the addition of bromide or pyridinium hydrobromide perbromide and the formation of an iodine derivative. Although these postcolumn derivatization approaches produce molecules that are more fluorescent than their precursors, the use of bromine or iodine requires extra pumps and chemical reactors for the HPLC system and it takes a long time to prepare the mobile phase. The use of postcolumn photochemical reactors is a novel derivatization methodology where the outlet of the HPLC is simply connected to ultraviolet permeable polytetrafluoroethylene tubing and wrapped over a high-intensity UV lamp. Stable and highly fluorescent derivatives are yielded from the reaction of mycotoxins with hydroxyl radicals from water, generated from the UV light irradiation. This alternative technique is simple, the response is linear, it has reproducibility and it does not require chemical reagents, additional pumps or electrochemical cells, and therefore it is more economical than the conventional postcolumn derivatization [82]. Lee et al. [141] applied photochemical derivatization to enhance AFs, OTA, and ZEN fluorescence in feed, Ok et al. [142] used it to increase this property in AFs present in corn, and Wu et al. [97] applied it to detect AFB1 in feed and raw feed ingredients (Table A7). 

Recently, the use of HPLC-DAD techniques has increased but they are incapable of dealing with a large number of analytes in complicated samples [82]. This technique was used to quantify DON and ZEN in wheat [76], DON and 3-AcDON in corn and feed [143], and DON in wheat and their by-products [144,145] (Table A7).

The UHPLC/UPLC methods have been newly introduced. Columns filled with uniform particles of small size and instruments with high-pressure fluidic modules are used. This rising technique allows decreased run times and solvent consumptions, resulting in more efficient chromatographic separations with higher sensitivity and resolution [57,82,99]. UHPLC/UPLC was explored by several authors to detect mycotoxins in feed and raw ingredients for feed (Table A7 and Table A8).

LC can be coupled to MS (LC-MS) or to MS/MS, which occurs via atmospheric pressure ionization (API) techniques such as electrospray ionization (ESI), atmospheric pressure chemical ionization (APCI), and atmospheric pressure photoionization (APPI). This has resulted in a very versatile analytical tool whose applications include not only single mycotoxin analysis, but, most importantly, true multi-mycotoxin determination. This is a current trend in this field since commodities can be contaminated with more than one mycotoxin, as discussed earlier.

Relatively to API methods, ESI is mostly well suited for the analysis of polar compounds. APPI is highly effective for the analysis of medium- and low-polar substances and APCI is often more sensitive than the majority of polar functional groups which are of moderate polarity [57,146]. Normally, as a consequence of API, protonated or deprotonated molecules can be produced [57]. With respect to ESI and mycotoxins, the protonated precursor ions are mainly formed, but additional information can be found in [147,148,149,150]. In Table A8, LC-MS methods are reviewed and it can be seen that the use of ESI interface is predominant in multi-mycotoxins applications. However, APCI and APPI interfaces usually have better performances in terms of chemical noise and signal suppression than ESI, despite being less used [146]. Normally, APCI is applied only to mycotoxins of the TRCs group, although its feasibility has also been examined in a few multi-mycotoxin methods [57]. Actually, Hofgaard et al. [151] employed this interface to quantify not only TRCs, but also ZEN and FBs in wheat. Nowadays, most of the instruments offer combined interfaces (ESI/APCI) which have a compromised sensitivity between both modes, however, offer the main advantage of enabling the detection of polar and nonpolar analytes in a single run [57]. In LC-MS/MS, the ionization process may have some problems and the analytical signal is unpredictable and it is affected by the matrix effects. Therefore, the use of isotope-labelled internal standards that are not naturally occurring in the samples and have identical chemical properties to the analytes will compensate for both losses during the sample pretreatment steps and for ion suppression or enhancement effects in the ion source. Despite being the best approach, these standards are only available for a limited number of mycotoxins and are very expensive [57,152].

The LC system can be combined with a single quadrupole, an ion trap (IT), a triple quadrupole (QqQ), or with a hybrid quadrupole/linear ion trap detector (QTRAP) [57,153]. The LC-MS/MS is enabled by QqQ or QTRAP [146]. As can be seen in Table A8, QqQ instruments by far surpassed by remaining analyzers, perhaps, due to improved signal to noise ratios from the additional selectivity of the second MS step [146]. In this field of analysis, IT was only used to detect multi-mycotoxins in finished feed, maize, and maize silage [43].

HRMS can be performed using time-of-flight (TOF) and Orbitrap analyzers, that have a high mass accuracy, high resolving power, and high dynamic range [57,104]. Even when these instruments are operating in full scan mode they are able to provide high sensitivity, which makes the identification of analytes easier even when they are present at very low levels. Additionally, they have rapid spectral acquisition speed that allow them to record virtually an unlimited number of compounds. Between the authors that chose these detectors (Table A8), TOF was more frequently applied than Orbitrap, despite knowing the advantage of this last detector to screen unknown compounds in full scan mode, in parallel to the quantification of known analytes [154]. 

Relying on the strengths of the exceptional sensitivity and separation capabilities of modern LC-MS equipment, “dilute and shoot” (DaS) methods have been developed [87]. They rely on sample dilution followed by a direct injection and they avoid a cleanup stage, which limits the potential loss of analytes. This is a rapid method that covers a wide range of polarities, and therefore allows a wide range of mycotoxins and other secondary metabolites to be determined. On the negative side, DaS has the risk of having excessive and unpredictable interference from the matrix which is a limitation as it can potentially overwhelm the sensitivity of the instrument [82,87,104,155]. 

## 6. Final Considerations

The world demand for commodities commonly used in the manufacture of animal feed, such as maize, wheat, and soybeans has been steadily rising in the last years, driven by higher demands for livestock production. This has led to an increased awareness of animal feed safety issues due to the fact that feed consumption is a potential route for chemical hazards to enter the human food chain. Within these hazards, mycotoxins deserve some prominence and AFs, FMs, OTs, TRCs, and ZEN are the most prevalent and worrying classes of compounds.

Mycotoxins represent a serious threat to the feed supply chain, animal health, and, in the limit, human health. So, regulatory agencies established limits to keep their levels in animal feeds under control. In this way, the protection of all parts likely affected by the presence of these toxins is somehow assured. The legislation (regulation or recommendation) applicable in the EU to products intended for livestock feed is very strict and can block exportation of feed commodities from developing countries to their European trading partners. A verified limitation in the legislation on mycotoxins is the fact that it does not consider the frequently reported and worrying scenario of multi-mycotoxin contamination of single commodities and animal feed.

The review of published reports from 2016 to 2018 on contamination of maize, wheat, soybeans, their by-products, and animal feed with legislated mycotoxins and their metabolites, made us realize that this is an issue that is increasingly relevant. In general, it was verified that the common association of maize with AFs and FMs, and of wheat with DON, favored the investigation of these mycotoxins. However, mycotoxin formation is a complex and multifactor phenomenon whose worldwide contamination and distribution patterns are predicted to be significantly affected by climate change because of the appearance of favorable environmental conditions for fungal proliferation in uncommon places. Therefore, the presence of mycotoxins is unpredictable, and therefore multi-mycotoxins surveys end become more realistic and preferred, since the study of only some of these contaminants provides insufficient information about the risks associated with a respective feedstuff. In addition, since co-occurrence was commonly reported in the years under review, it is expected that this phenomenon will be further addressed in the coming years. Specifically, regarding soybean and their by-products, they are less targeted as compared with other matrices because these fungal toxins are not considered to be very problematic in this commodity. 

With respect to testing methods, in the future, it is expected that there will be an expansion of sample pretreatment techniques that are aimed at the minimization and automation of these procedures, although classical methods like SLE will probably still be applied prior to some detection approaches, as verified in this review. Concerning LC, similar to what happened in last years, the use of the HPLC and LC-MS methodologies to quantify mycotoxins in animal feed, will perhaps continue side-by-side. Furthermore, detection methodologies that target several mycotoxins will surely gain ground, and, probably, developments will occur in screening methods that allow analysis in the field.

Finally, in our point of view, the mycotoxins field of analysis within the matrices in review is not expected to decline and the industries of animal production systems will become even more aware of the relevance of these contaminants in order to improve the quality and safety of products intended for animal feed.

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
