# Peer review of "Prevalent Mycotoxins in Animal Feed: Occurrence and Analytical Methods"

_toxins, 2019, doi:10.3390/toxins11050290_

Reviewer 1 Report

This paper reviews the occurrence and the recent advances of analytical methods for mycotoxins in animal feed. This topic would be attractive to a broad readership in the field of mycotoxins. The review was organized clearly, and the manuscript was written very well. It is suitable for publication after minor revisions.

1. Line 186, "mycotoxins occurrence" should be "mycotoxin occurrence". Please check the manuscript and revise like this.
2. Line 317, "Mycotoxin determination methods" should be better.
3. Line 455, "Combined extraction/clean-up/concentration" would be better.
4. Line 889, Table 3, Please specify the columns which were used in HPLC studies.
5. Line 893, Table 4, not sure if MS/MS was used for all the studies in the tables. Writing just "LC-MS" should be good. 

Author Response

Dear reviewers

We would like to thank the reviewers for the careful and thorough reading of this paper and for the thoughtful comments and constructive suggestions, which helped to improve the quality of our manuscript Toxins-507015 “Prevalent mycotoxins in animal feed: occurrence and analytical methods”.

The response follows (in Italic).

Reviewer 1

This paper reviews the occurrence and the recent advances of analytical methods for mycotoxins in animal feed. This topic would be attractive to a broad readership in the field of mycotoxins. The review was organized clearly, and the manuscript was written very well. It is suitable for publication after minor revisions.

1.            Line 186, "mycotoxins occurrence" should be "mycotoxin occurrence". Please check the manuscript and revise like this.

The subtitle was corrected as suggested.

2. Line 317, "Mycotoxin determination methods" should be better.

The title was corrected as suggested.

3. Line 455, "Combined extraction/clean-up/concentration" would be better.

The subtitle was corrected as suggested.

4. Line 889, Table 3, Please specify the columns which were used in HPLC studies.

The columns were added as suggested in Table 3.

5. Line 893, Table 4, not sure if MS/MS was used for all the studies in the tables. Writing just "LC-MS" should be good.

The correction was made as suggested.

Reviewer 2 Report

TOXINS

Prevalent mycotoxins in animal feed: occurrence and analytical methods

The topic of the manuscript is interesting and of high scientific impact on the field, only minor issues should be clarified before acceptance for publication.

Line 74 pag 2. Add Alternaria to the genus Aspergillus, Penicillium, and Fusarium

Line 84: A. nomius in italic.

Line 740 pag 16. The sentence “because their manipulation is easier and aqueous mobile phases are less toxic” is not very appropriate; in my opinion, the great majority of separations are performed on reverse-phase columns because the majority of mycotoxins are soluble in polar organic solvents as methanol, acetonitrile, water and in their mixtures.

Author Response

Dear reviewers

We would like to thank the reviewers for the careful and thorough reading of this paper and for the thoughtful comments and constructive suggestions, which helped to improve the quality of our manuscript Toxins-507015 “Prevalent mycotoxins in animal feed: occurrence and analytical methods”.

The response follows (in Italic).

Reviewer 2

Line 74 pag 2. Add Alternaria to the genus Aspergillus, Penicillium, and Fusarium

The genus was added as suggested

Line 84: A. nomius in italic.

The italic was used as suggested.

Line 740 pag 16. The sentence “because their manipulation is easier and aqueous mobile phases are less toxic” is not very appropriate; in my opinion, the great majority of separations are performed on reverse-phase columns because the majority of mycotoxins are soluble in polar organic solvents as methanol, acetonitrile, water and in their mixtures.

The sentence was corrected as suggested.